# Influence of Physical Activity during Pregnancy on Neonatal Complications: Systematic Review and Meta-Analysis

**DOI:** 10.3390/jpm14010006

**Published:** 2023-12-20

**Authors:** Cristina Silva-Jose, Linda May, Miguel Sánchez-Polán, Dingfeng Zhang, Alejandro Barrera-Garcimartín, Ignacio Refoyo, Rubén Barakat

**Affiliations:** 1AFIPE Research Group, Faculty of Physical Activity and Sport Sciences-INEF, Universidad Politécnica de Madrid, 28040 Madrid, Spain; cristina.silva.jose@upm.es (C.S.-J.); miguelsanpol@gmail.com (M.S.-P.); zhangdingfeng123@gmail.com (D.Z.); a.barrerag@alumnos.upm.es (A.B.-G.); 2Department of Kinesiology, East Carolina University, Greenville, NC 27834, USA; mayl@ecu.edu; 3Sports Department, Faculty of Physical Activity and Sports Sciences-INEF, Universidad Politécnica de Madrid, 28040 Madrid, Spain; ignacio.refoyo@upm.es

**Keywords:** pregnancy, physical activity, NICU, neonatal, complications

## Abstract

Newborn hospitalisations after delivery are indicators of poor neonatal health with potential risks of future diseases for children. Interventions to promote a healthy environment have been used during pregnancy, with physical activity as a principal element. A systematic review and meta-analyses were performed to evaluate the effect of physical activity during pregnancy on neonatal intensive care unit (NICU) admissions and Apgar 1 and 5 scores (Registration No.: CRD42022372493). Fifty studies (11,492 pregnant women) were included. There were significantly different rates of NICU admissions between groups (RR = 0.76, 95% CI = 0.62, 0.93; Z = 2.65, *p* = 0.008; I2 = 0%, and *P*_heterogeneity_ = 0.78), and significant differences in Apgar 1 (Z = 2.04; *p* = 0.04) (MD = 0.08, 95% CI = 0.00, 0.17, I2 = 65%, *P*_heterogeneity_ = 0.00001) and Apgar 5 (Z = 3.15; *p* = 0.002) (MD = 0.09, 95% CI = 0.04, 0.15, I2 = 80%, and *P*_heterogeneity_ = 0.00001), favouring intervention groups. Physical activity during pregnancy could help to reduce the risk of NICU admissions that are related to neonatal complications.

## 1. Introduction

Evidence of poor neonatal health includes the occurrence of newborn hospitalisations after delivery [1]. The consequences of an unhealthy pregnancy and associated conditions, such as gestational diabetes or hypertension, translate into short-term risks for the newborn that result in long-term consequences for the infant [2]. Furthermore, hypothalamic-pituitary-adrenal activation throughout pregnancy is linked to fetal programming [3].

The admission to the neonatal intensive care unit (NICU) after delivery is associated with neonates born prematurely and/or growth-restricted (i.e., small-for-gestational age and intrauterine growth restriction). In addition to IUGR and premature delivery, NICU admissions are also related to low Apgar scores, excess neonatal body fat (i.e., high body weight, ponderal index, and BMI), low cord blood pH, hyperbilirubinemia, neonatal hypoglycemia (serum glucose concentration < 40 mg/dL), shoulder dystocia, and brachial plexus injury [4,5,6]. Other common complications of neonates in the NICU are hyperglycemia (serum glucose concentration > 150 mg/dL), as well as respiratory and metabolic disorders in infancy [7,8]. Additionally, neonates with hyperglycemia are often born with extremes in birth weight, either too small or too large [1].

Further, these fetal and neonatal complications, which lead to NICU admission, are associated with long-term health problems [9,10]. For example, short-term neonatal complications soon after delivery are associated with negative long-term effects on childhood development (e.g., low neurodevelopment scores) [11] and health across the lifespan (e.g., obesity, diabetes, and cardiovascular disease) [12,13]. Furthermore, high anxiety in mothers, autonomously of the baby’s birth situation and the moment of evaluation, comprises a possible risk factor for the child’s development, even in the fetal period [14].

Conversely, physical activity during pregnancy, meeting the international recommendations of 150 min of moderate physical activity every week, benefits the fetal-placental-maternal interface [15]. Therefore, access of pregnant women to physical activity is vital for promoting the short- and long-term health of the child. However, whether maternal physical activity during pregnancy impacts the incidence of neonatal complications is not known [16,17,18].

Despite many known benefits for the mother, prenatal exercise was previously thought to increase the odds of neonatal complications, such as preterm birth and intrauterine growth restriction [12]. Conversely, studies show that admission to the NICU admission for infants is less common in women who exercise during pregnancy [5].

Furthermore, research has shown that physical activity (PA) during pregnancy could have a positive effect on neonatal and maternal outcomes, which is closely linked to NICU admissions. For instance, PA during pregnancy decreases gestational weight gain and the risk of gestational diabetes while increasing gestational age and Apgar scores [19,20].

PA is seen as an enhancer of maternal and fetal health, improving these interrelated health outcomes. Moreover, the association between the time spent in physical activity and symptoms of depression and anxiety in the antenatal period was previously demonstrated [21]. Unfortunately, only 20% of the world population meets the international guidelines for exercise during pregnancy [22]. Further, the worldwide prevalence of pregnant women who do not meet the ACOG physical activity recommendations is steadily rising [23]. Since the influence of PA during pregnancy on neonatal outcomes remains unclear [24], it is necessary to review the effect of maternal physical activity during pregnancy on neonatal health. Therefore, the purpose of this review is to examine the correlations between physical activity during pregnancy and NICU admission and Apgar scores after delivery.

## 2. Materials and Methods

This systematic review was completed using the Preferred Reporting Items for Systematic Reviews and Meta-Analyses (P.R.I.S.M.A.) guidelines. The protocol was registered in the International Prospective Registry of Systematic Reviews (PROSPERO) (CRD42022372493).

### 2.1. Eligibility Criteria

The PICOS (population, intervention, comparison, outcome, and study design) strategy was used to guide this review with meta-analysis [25].

#### 2.1.1. Population

The population of interest comprised of healthy pregnant women aged between 18 and 45 years, regardless of their gestational age at the time of study admission. Participants had no contraindications for physical activity or exercise, as defined by the American College of Obstetricians and Gynecologists (ACOG) guidelines [26].

Absolute contraindications were defined as follows: placenta previa, premature labour, multifetal pregnancies, persistent second or third-trimester bleeding, incompetent cervix, intrauterine growth restriction, ruptured membranes, serious cardiovascular, respiratory, or systemic disorders [26].

#### 2.1.2. Intervention

Interventions that include any format of physical activity or physical exercise during pregnancy (individual/group), (autonomous/supervised), (face-to-face/online), and co-interventions: exercise combined with other interventions (e.g., diet intervention or behavioral intervention) were evaluated. The intervention investigated in each study must be related to objective or subjective measures of intensity, duration, volume, or type of exercise.

#### 2.1.3. Comparison

In this case, the comparison group was the control or non-intervention group of the selected studies. This was based on not practicing routine physical activity during pregnancy, following only standard medical care.

#### 2.1.4. Variable

The primary study variable was the NICU admissions record. The studies had to contain at least the primary study variable (NICU admission) to be included in the analysis; if a study failed this step, then it was registered as a potentially relevant secondary variable for analysis. The secondary variable was Apgar 1 and Apgar 5 scores in quantitative and qualitative format.

#### 2.1.5. Study Design

Studies that were randomized clinical trial interventions were selected. Thus, studies with non-randomized interventions, observational studies, some type of review (narrative, systematic, or systematic review with meta-analysis), and qualitative research were excluded.

### 2.2. Data Sources

An exhaustive and comprehensive search was completed in the following databases: MEDLINE, Scopus, Sport Discus, Academic Search Premier, ERIC, OpenDissertations, Clinicaltrial.gov, Cochrane Database of Systematic Reviews, and Web of Science through the portal of the Universidad Politécnica de Madrid.

The search was conducted between October 2022 and November 2022. The same article selection criteria were used for all the databases to guarantee equality. In the selection process, articles written in Spanish and English published between 2010 and 2023 were considered for the search. Bibliographic references of selected studies were reviewed to identify other potential studies that might have been missed by the electronic keyword search.

### 2.3. Selection and Data Extraction

Figure 1 shows the selection process that was followed for the reviewed articles. Two investigators independently screened the titles and abstracts identified from the electronic searches to select potentially relevant studies based on the inclusion criteria.

Abstracts were identified and passed an initial screen; then, full-text searches were performed post hoc. Full texts were reviewed separately for priority results for data extraction. In addition, relevant data were extracted from 50 studies to ensure that no valuable information was missed.

For studies in which one of the authors recommended exclusion, both authors reached a consensus to make a final decision on inclusion or not. In situations of absolute disagreement, a third author provided their assessment of study inclusion or not. One person extracted the data to complete the tables, which were independently verified by a content expert to facilitate further analysis.

In this review, data were extracted from tables or from the text using simple methods, excluding articles that presented data in figure form. This way, the reliability and authenticity of the data are guaranteed. Extracted data included study characteristics (e.g., author’s last name and year, country), article type (randomized clinical trial—RCT), sample size and group differences, intervention/exposure (exercise prescribed and/or measured), including frequency, intensity, time, and type, supervision of the intervention, duration, and adherence to the intervention). Primary and secondary variable(s) were analysed, and co-intervention associations, if any, were also considered (Table 1).

### 2.4. Evidence Quality Assessment

To assess the quality of evidence for each study design and main outcome, the Grading of Recommendations, Assessment, Development, and Evaluation (GRADE) framework was used. This framework provided a standardized and comprehensive approach to assess the strength of evidence across multiple studies [27]. There were 50 randomized clinical trials rated as high included in this meta-analysis.

### 2.5. Risk of Bias Assessment

The Cochrane Handbook was used to assess the risk of bias (Figure 2). Potential sources of bias were assessed in all studies. These sources included selection bias (inadequate randomisation procedures for RCTs/interventions), performance bias (intervention compliance for RCTs/interventions), detection bias (faulty outcome measurement), attrition bias (incomplete follow-up and high loss of participants during follow-up), and reporting bias (incomplete or selective reporting of results) [28]. Overall, the quality of the evidence ranged from low to high. Although the risk of bias was detected, it was chosen not to exclude any study from the analyses.

### 2.6. Publication Bias Assessment

To assess potential publication bias in each developed meta-analysis, the Egger regression test was employed due to its enhanced sensitivity in detecting publication bias under conditions of weak or moderate heterogeneity. Typically, this test yields a metric indicating significant publication bias when *p* < 0.1 [29].

### 2.7. Statistical Analysis

Statistical analyses were performed using RevMan (version 5.4) software. NICU admission and Apgar 1 > 7 and Apgar 5 > 7 were expressed as dichotomous categorical outcomes (Yes/No); the number of NICU admissions and Apgar 1 > 7 and Apgar 5 > 7 events in the intervention and control groups and their relative risk (RR) were recorded. A random-effects model was used to calculate the total sum of the RR [30]. To establish the balanced mean in the dichotomous analysis, a weighting system was used that considered the sample size or number of events reported by each study. This weighting system accounts for the different levels of information provided by each study, which allows a more precise representation of the general data. The I^2^ statistic was used to measure the degree of heterogeneity of the results. This metric provides information about the proportion of variability in the observed intervention effect between studies that is attributed to heterogeneity rather than chance. The I^2^ statistic was interpreted using established thresholds: 25% for low heterogeneity, 50% for moderate heterogeneity, and >75% for high heterogeneity [30]. For continuous outcomes, Apgar 1 and Apgar 5 scores, obtained through medical records of analysed articles, the standardized mean difference (SMD) was used. The overall confidence interval (CI) was calculated using the standardized mean difference [30].

**Figure 2 jpm-14-00006-f002:**
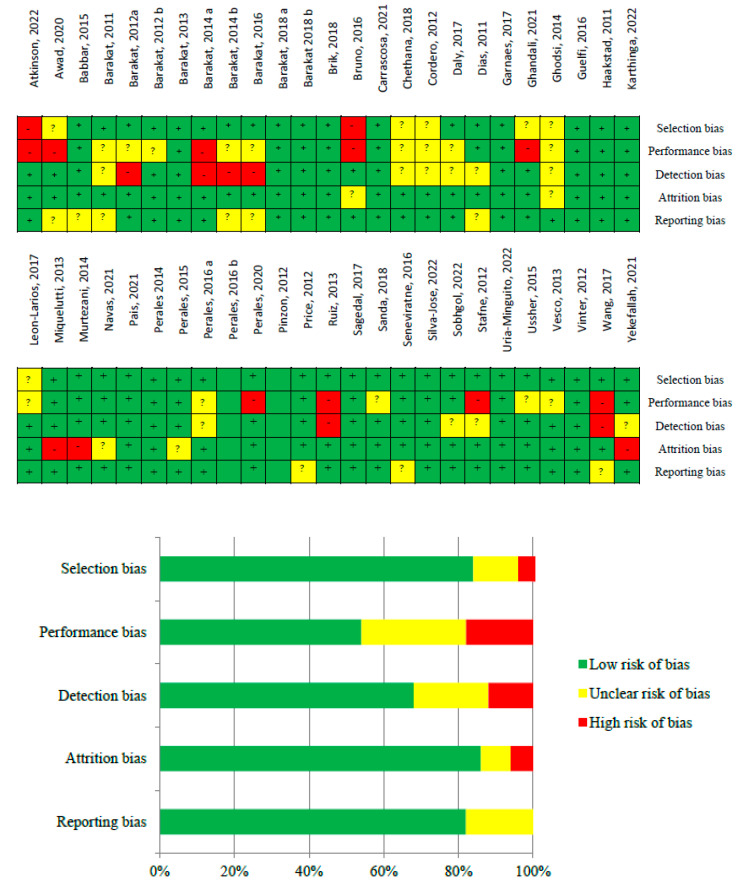
Risk of bias of selected studies [5,31,32,33,34,35,36,37,38,39,40,41,42,43,44,45,46,47,48,49,50,51,52,53,54,55,56,57,58,59,60,61,62,63,64,65,66,67,68,69,70,71,72,73,74,75,76,77,78,79].

**Table 1 jpm-14-00006-t001:** Study characteristics.

Ref	Autor	Year	Country	Type	N	IG	CG	Intervention. Exercise Program							Principal Outcomes	Secondary Outcomes	Co-Intervention
								Freq	Int	Tp	Type	Sup.	Dur.	Adh.			
[31]	Atkinson et al.	2022	Norway	RCT	331	164	117	3	Moderate	12 wk	Strength Endurance Balance	Supervised	60	ND	Child height and weight, BMI, and physical activity,	Maternal outcomes and neonatal outcomes	No
[32]	Awad et al.	2020	Egypt	RCT	50	25	25	3 + 3	Moderate	22 wk	Aerobic and PFMT	Mix	60 + 35	ND	Duration of labour	Neonatal outcomes	No
[33]	Babbar et al.	2016	USA	RCT	46	23	23	3	Moderate	8 wk	Yoga	Supervised	60	80%	Birth weight and type of delivery	Maternal outcomes and neonatal outcomes	No
[34]	Barakat et al.	2011	Spain	RCT	80	40	40	3	Moderate	28 wk	Aerobic and strength exercises	Supervised	35–45	ND	Maternal health status	Maternal outcomes and neonatal outcomes	No
[35]	Barakat et al.	2012a	Spain	RCT	290	138	152	3	Moderate	28 wk	Aerobic exercises	Supervised	40–45	ND	Type of delivery	Maternal outcomes and neonatal outcomes	No
[36]	Barakat et al.	2012b	Spain	RCT	83	40	43	3	Moderate	28 wk	Land and aquatic exercises	Supervised	35–45	80%	Gestational weight gain and 37 gestational diabetes	Maternal outcomes and neonatal outcomes	No
[37]	Barakat et al.	2013	Spain	RCT	510	255	255	3	Moderate	28 wk	Aerobic, strength and flexibility	Supervised	50–55	95%	Gestational diabetes	Maternal outcomes and neonatal outcomes	No
[38]	Barakat et al.	2014a	Spain	RCT	200	107	93	3	Moderate	28 wk	Aerobic exercises and PFMT	Supervised	55–60	80%	Gestational weight gain and type of delivery	Maternal outcomes and neonatal outcomes	No
[39]	Barakat et al.	2014b	Spain	RCT	290	138	152	3	Moderate	28–31 wk	Aerobic exercises	Supervised	55–60	80%	Gestational age	Maternal outcomes and neonatal outcomes	No
[40]	Barakat et al.	2016	Spain	RCT	765	382	383	3	Moderate	28 wk	Aerobic, strength, and flexibility	Supervised	50–55	80%	Hypertension	Maternal outcomes and neonatal outcomes	No
[41]	Barakat et al.	2018a	Spain	RCT	429	227	202	3	Moderate	28 wk	Aerobic exercises, PFMT, and flexibility	Supervised	55–60	85%	Duration of labour	Maternal outcomes and neonatal outcomes	No
[42]	Barakat et al.	2018b	Spain	RCT	65	33	33	3	Moderate	28 wk	Aerobic exercises	Supervised	55–60	ND	Placenta weight	Maternal outcomes and neonatal outcomes	No
[43]	Brik et al.	2019	Spain	RCT	85	42	43	3	Moderate	30 wk	Aerobic, strength, coordination, balance, PFMT, and stretching and relaxation.	Supervised	60	70%	Maternal weight during pregnancy	Fetal and neonatal outcomes.	No
[44]	Bruno et al.	2016	Italy	RCT	131	69	62	3	Moderate	30 wk	Physical activity recommendations by the ACOG and the ACSM	Not Supervised	30	ND	Gestational Diabetes	Maternal outcomes and neonatal outcomes	Diet
[45]	Carrascosa et al.	2021	Spain	RCT	286	145	141	3	Moderate	20 wk	Aquatic aerobic exercises	Supervised	45	ND	Epidural and analgesia during labour	Maternal outcomes and neonatal outcomes	No
[46]	Chetana et al.	2018	India	RCT	150	75	75	3	Moderate	7 wk	Yoga	Supervised	30	ND	Labour pain intensity	Fetal and neonatal outcomes	No
[47]	Cordero et al.	2015	Spain	RCT	257	101	156	1 + 2	Low	26 wk	Land aerobics and aquatic activity	Supervised	50 + 60	80%	Gestational diabetes	Maternal outcomes and neonatal outcomes	No
[48]	Daly et al.	2017	Ireland	RCT	86	43	43	3	Moderate	28 wk	Aerobic, strength, and PFMT	Supervised	50–60	78.9%	Fasting plasma glucose	Maternal outcomes and neonatal outcomes	No
[49]	Dias et al.	2011	Norway	RCT	42	21	21	1 + 6	Low	16 wk	PFMT	Mix	30	75%	Type of delivery	Maternal outcomes and neonatal outcomes	No
[50]	Garnæs et al.	2016	Norway	RCT	91	46	45	3	Moderate	20–26 wk	Treadmill walking, strength training, and PFMT	Supervised	60	50%	Gestational Weight Gain	Maternal outcomes and neonatal outcomes	No
[51]	Ghandali et al.	2021	Iran	RCT	103	51	52	2	Low–moderate	8 wk	Pilates	Supervised	35	ND	Type of delivery	Maternal outcomes and neonatal outcomes	No
[52]	Ghodsi et al.	2014	Iran	RCT	80	40	40	3	Low	15 wk	Stationary cycling	No supervised	15	ND	Gestational weight gain	Maternal outcomes and neonatal outcomes	No
[53]	Guelfi et al.	2016	Australia	RCT	172	85	87	3	Moderate	14 wk	Stationary cycling	Supervised	20–60	79%	Diagnosis of GDM	Maternal outcomes and neonatal outcomes	No
[54]	Haakstad et al.	2020	Norway	RCT	105	52	53	2 + 1	Moderate	12 wk	Aerobic dance and strength	Mix	60 + 30	80%	Birth weight	Maternal outcomes and neonatal outcomes	No
[55]	Karthiga et al.	2022	India	RCT	234	121	113	2	low	20 wk	yoga	Not supervised	60	ND	Gestational hypertension	Maternal outcomes and neonatal outcomes	No
[56]	Leon-Larios et al.	2017	Spain	RCT	466	254	212	5	Low	6 wk	PFMT	Not supervised	18–23	ND	Perineal tear and episiotomy	Maternal outcomes and neonatal outcomes	Perineal massage
[57]	Miquelutti et al.	2013	Brazil	RCT	149	78	71	7	Low	14 wk	Aerobic and PFMT	Not supervised	10–30	ND	Urinary incontinence	Maternal outcomes and neonatal outcomes	No
[58]	Murtezani et al.	2014	Kosovo	RCT	63	30	33	3	Moderate	20 wk	Aerobic and strength exercises	Supervised	40–45	85%	Birth weight	Maternal outcomes and neonatal outcomes	No
[59]	Navas	2021	Spain	RCT	320	148	146	3	Moderate	20 wk	Aquatic, PFMT, and breathing and relaxation exercises	Supervised	45 min	ND	Postpartum depression, sleep problems, and maternal quality of life	Maternal outcomes and neonatal outcomes	No
[60]	Pais et al.	2021	India	RCT	124	61	63	7	Moderate	1 wk	Yoga	Supervised	45	ND	Incidence of preeclampsia and preterm birth	Maternal, fetal and neonatal outcomes	No
[61]	Perales et al.	2014	Spain	RCT	167	90	77	3	Moderate	29 wk	Aerobic exercises	Supervised	55–60	ND	Prenatal depression	Maternal outcomes and neonatal outcomes	No
[62]	Perales et al.	2015	Spain	RCT	63	38	25	3	Moderate	28 wk	Aerobic dance and PFMT	Supervised	55–60	80%	Fetal and maternal heart rate	Maternal outcomes and neonatal outcomes	No
[63]	Perales et al.	2016a	Spain	RCT	166	83	83	3	Low–moderate	28 wk	Aerobic, strength, and PFMT	Supervised	55–60	ND	Duration of labour	Maternal outcomes and neonatal outcomes	No
[64]	Perales et al.	2016b	Spain	RCT	241	121	120	3	Low–moderate	28 wk	Aerobic and strength exercises	Supervised	55–60	70%	Maternal cardiovascular health	Maternal outcomes and neonatal outcomes	No
[65]	Perales et al.	2020	Spain	RCT	1348	668	660	3	Low-moderate	30 wk	Aerobic and PFMT	Supervised	50–55	ND	Gestational weight gain	Maternal outcomes and neonatal outcomes	No
[66]	Pinzon et al.	2012	Colombia	RCT	64	31	33	3	Moderate	12 wk	Aerobic and flexibility exercises	Supervised	60	ND	Gestational weight gain	Maternal outcomes and neonatal outcomes	No
[67]	Price et al.	2012	USA	RCT	62	31	31	3 + 1	Moderate	23 wk	Aerobic and walk	Mix	30 + 60	ND	Gestational weight gain	Maternal outcomes and neonatal outcomes	No
[68]	Ruiz et al.	2013	Spain	RCT	962	481	481	3	Low–Moderate	28 wk	Aerobic and strength exercises	Supervised	50–55	97%	Gestational weight gain	Maternal outcomes and neonatal outcomes	No
[69]	Sagedal et al.	2017	Norway	RCT	591	296	295	2	Moderate	24 wk	Strength training and cardiovascular exercise	Supervised	60	ND	Gestational weight gain	Maternal outcomes and neonatal outcomes	Diet
[70]	Sanda et al.	2018	Norway	RCT	589	295	294	2 + 3	Moderate	24 wk	Aerobic, strength exercises, and PFMT	Mix	50	ND	Duration and mode of delivery	Maternal outcomes and neonatal outcomes	No
[71]	Seneviratne et al.	2015	New Zealand	RCT	75	38	37	3–5	Moderate	16 wk	Aerobic exercises	Supervised	15–30	33%	Birth weight	Maternal outcomes and neonatal outcomes	No
[5]	Silva-José et al.	2022	Spain	RCT	139	69	70	3	Moderate	30 wk	aerobic exercise, strength, balance and coordination, and PFMT and flexibility	Supervised	55–60	80%	Birth weight	Maternal outcomes and neonatal outcomes	No
[72]	Sobhgol et al.	2022	Australia	RCT	200	100	100	1–2	Low	16 wk	PFMT	Not supervised	10	50%	Female sexual function	Maternal outcomes and neonatal outcomes	No
[73]	Stafne et al.	2012	Norway	RCT	702	375	327	3	Moderate–High	14–16 wk	Aerobic activity, strength training, and balance exercises.	Supervised	60	55%.	Gestational diabetes	Maternal outcomes and neonatal outcomes	No
[74]	Uria-Minguito et al.	2022	Spain	RCT	203	102	101	3	Moderate	28 wk	Aerobic, strength, balance, coordination, PFMT, and flexibility	Supervised	50–60	ND	Gestational diabetes	Maternal outcomes and neonatal outcomes	No
[75]	Ussher et al.	2015	England	RCT	774	384	391	2 + 1	Moderate	8 wk	Treadmill and walking	Supervised	30	40%	Smoking cessation	Maternal outcomes and neonatal outcomes	No
[76]	Vesco et al.	2013	USA	RCT	114	56	58	7	Moderate	ND	Physical activity recommendations	Not supervised	30	ND	Gestational weight gain	Maternal outcomes and neonatal outcomes	Diet
[77]	Vinter et al.	2012	Denmark	RCT	304	150	154	7	Moderate	21 wk	Walking exercises	Not supervised	30–60	ND	Gestational weight gain	Maternal outcomes and neonatal outcomes	Diet
[78]	Wang et al.	2017	China	RCT	226	112	114	3	Moderate	24 wk	Stationary cycling	Supervised	45–60	75%	Gestational diabetes	Maternal outcomes and neonatal outcomes	No
[79]	Yekefallah et al.	2021	Iran	RCT	70	35	35	3	Low–moderate	11 wk	Yoga	Supervised	75	ND	Episiotomy and perineal tear	Maternal outcomes and neonatal outcomes	No

Ref: references. Author: last name. Year: year of study. Country: the country where the article has been completed. Type: type of article; N: total number of women analysed. GI: number of women analysed in the intervention group. GC: number of women analysed in the control group. Freq: weekly frequency of exercise sessions. Intens: type of intensity. Tp: program time. Type: type of exercise performed. Superv. classes: whether or not there was supervision. Duration: minutes of each session. Adh.: adherence of the participants to the intervention (%). Main variables and secondary variables: the same as before but secondary. Wk: week.

## 3. Results

### 3.1. Study Selection

The PRISMA diagram presenting the search results along with explanations for the study exclusion is shown in Figure 1.

### 3.2. Study Characteristics

Altogether, 50 studies involving 11,492 pregnant women in 15 countries on 5 continents met the inclusion criteria. Of the 50 studies, 34 were conducted in Europe, specifically in Spain [5,34,35,36,37,38,39,40,41,42,43,45,47,56,59,61,62,63,64,65,68,74], Norway [31,49,54,60,69,70,73], Italy [44], Ireland [48], England [75], Kosovo* [58], and Denmark [77]. The remaining 16 were performed in other countries worldwide—seven in Asia: India [46,55,60], Iran [51,52,79], and China [78], five in America: United States [33,67,76], Brazil [57], and Colombia [66], three in Oceania: Australia [53,72] and New Zealand [71], and one in Africa: Egypt [32].

All studies were randomized clinical trials, with 45 having exercise interventions only, 4 with dietary counselling co-interventions [44,69,76,77], and 1 with perineal massages [56].

Specifically, the exercise interventions were completed in a range of low to moderate intensity, with a frequency varying from 1 to 7 days a week and a duration between 10 to 60 min per session. The interventions vary in their timing across gestation, with interventions throughout the pregnancy (week 8 to week 39) and other interventions during each trimester (week 8 to week 39). The type of exercise includes aerobics, resistance/muscular strength training, stretching, pelvic floor muscle training, balance, coordination, flexibility, or Pilates and yoga programs. Additional details are shown in Table 1.

Regarding the secondary variables, it was found maternal outcomes that include demographic characteristics (Age, parity, race, marital status, education, gravidity, maternal height and weight, hypertension, blood pressure, gestational diabetes, maternal fitness, psychological well-being, physical activity, and quality of life) and prenatal outcomes (gestational weight gain, gestational length of delivery, excessive weight gain, preeclampsia, mode of delivery, duration of labour, analgesia requirement during labour, episiotomy, and perineal tears). On the other hand, neonatal outcomes such as birth weight, low birth weight, small-for-gestational age, fetal growth restriction, fetal distress, preterm birth, placental weight, or neonatal death and other morbidities were recorded. NICU admission and Apgar Score results are presented in the following paragraphs.

### 3.3. Effect of Physical Activity during Pregnancy on NICU Admission

In this analysis, a total of 15 studies were included [5,43,44,46,48,50,53,55,60,69,71,73,75,76,77]. Participating in exercise during pregnancy leads to significantly different rates of NICU admissions (RR = 0.76, 95% CI = 0.62, 0.93; Z = 2.65, *p* = 0.008; I^2^ = 0%, *P_heterogeneity_* = 0.78). Figure 3 depicts the forest plot corresponding to the conducted meta-analysis. Quantification evaluation of the risk of publication bias test in the analysed articles showed that there was no potential publication bias (*p* = 0.63) in this analysis.

### 3.4. Effect of Physical Activity during Pregnancy on Apgar 1 > 7

In this qualitative analysis, a total of six randomized clinical trial studies were incorporated [40,48,51,52,55,57]. No statistical significance was found between physical activity during pregnancy and the occurrence of Apgar 1 > 7 (RR = 1.03, 95% CI = 0.98, 1.08; Z = 1.18, *p* = 0.24; I^2^ = 63%, *P_heterogeneity_* = 0.02). The quantitative assessment of publication bias risk in the analysed articles indicated the absence of potential publication bias (*p* = 0.1) in this analysis. Figure 4 illustrates the forest plot corresponding to the conducted meta-analysis.

### 3.5. Effect of Physical Activity during Pregnancy on Apgar 5 > 7

In this qualitative analysis, a total of 11 randomized clinical trial studies were incorporated [33,40,44,46,48,53,57,67,69,73,75]. No statistical significance was found between physical activity during pregnancy and the occurrence of Apgar 5 > 7 (RR= 1.00, 95% CI = 1.00, 1.01; Z = 1.18, *p* = 0.68; I^2^ = 0%, *P_heterogeneity_* = 0.63). The quantitative evaluation of publication bias risk in the analysed articles revealed no significant indication of potential publication bias (*p* = 0.28) in this analysis. Figure 5 depicts the forest plot corresponding to the conducted meta-analysis.

### 3.6. Effect of Physical Activity during Pregnancy on Apgar 1

In this quantitative analysis, a total of 33 randomized clinical trial studies were incorporated [5,31,32,34,35,36,37,38,39,41,42,45,47,49,50,51,54,56,58,59,61,62,63,64,65,66,67,68,71,72,74,78,79]. Significantly lower Apgar 1 values were found in the intervention groups compared to control groups (Z = 2.04; *p* = 0.04) (MD = 0.08, 95% CI = 0.00, 0.17, I^2^ = 65%, *P_heterogeneity_* = 0.00001). Figure 6 illustrates the forest plot corresponding to the conducted meta-analysis. The quantitative assessment of publication bias risk in the analysed articles revealed no significant indication of potential publication bias (*p* = 0.48) in this analysis.

### 3.7. Effect of Physical Activity during Pregnancy on Apgar 5

In this quantitative analysis, a total of 32 randomized clinical trial studies were incorporated [5,31,32,34,35,36,37,38,39,41,42,45,47,49,50,51,54,56,58,59,61,62,63,64,65,66,67,68,71,72,74,79]. Significantly lower Apgar 5 values were found in the control groups compared to intervention groups (Z = 3.15; *p* = 0.002) (MD = 0.09, 95% CI = 0.04, 0.15, I^2^ = 80%, *P_heterogeneity_* = 0.00001). Figure 7 depicts the forest plot corresponding to the conducted meta-analysis. The quantitative assessment of publication bias risk in the analysed articles showed no significant indication of potential publication bias (*p* = 0.75) in this analysis.

### 3.8. Effect of Physical Activity during Pregnancy on other Neonatal Outcomes

In addition, an attempt was made to analyse data from other present neonatal birth outcomes, such as shoulder dystocia, brachial plexus injury, cord blood pH, hyperbilirubinemia, or neonatal hypoglycaemia, but the analyses could not be performed due to their limited occurrence.

## 4. Discussion

According to our understanding, the present study is the first systematic review of the influence of regular physical activity on NICU admissions and Apgar scores, delving into 50 quality articles on said theme. This work provides a key contribution to support the idea that physical activity during pregnancy helps prevent admissions to the neonatal intensive care unit and improves APGAR scores. At this point, it was observed that maintaining physical activity during pregnancy increases the probability of having better postnatal outcomes and preventing complications.

Our systematic review with meta-analysis examined the relationship between systematic and regular physical activity during pregnancy and NICU admission and observed consistent findings compared to previous studies [80,81]. In this sense, lower NICU admissions have been found in the group of women who maintain regular physical activity; these results are consistent with previous studies that suggest newborns of women who exercised while pregnant were healthier. In this way, fewer neonatal complications are associated with babies who do not go to the NICU [5].

Up to this point, previous scientific evidence does not know whether maternal physical activity during pregnancy affects the incidence of neonatal complications [16,17,18]. What is known is that neonatal complications are associated with long-term negative effects on child development (e.g., neurodevelopmental scores) and lifelong health (e.g., obesity, diabetes, or cardiovascular diseases) [11,12,13]. Research has shown that physical activity during pregnancy could have a positive effect on neonatal outcomes, such as Apgar scores [19]; thus, encouraging physical activity during pregnancy by clinicians and inpatient institutions is vital to prevent delivery complications.

Consequently, pregnant women should lead a healthy lifestyle throughout pregnancy. To do this, study findings must be carefully and collectively weighed to formulate unique decisions for each woman’s situation. Additionally, more research is required to determine the lowest and maximum effects of physical activity levels throughout the entire pregnancy.

On the other hand, when it comes to physical exercise programs with pregnant women, PA is seen as an enhancer of maternal and fetal health, improving health parameters that may be interrelated [82]. For this review, studies were found with different objectives, methodologies, and interventions, although the central axis of these programs was strength and aerobic exercises. The plurality in the typology of interventions makes generalisation and firm extrapolation of results difficult. However, interventions with physical exercise as an agent to mitigate these negative effects are an urgent research approach.

It is important to consider that this systematic review and meta-analysis has several limitations in its approach, such as heterogeneity, publication bias, or the quality of the evidence. While the findings of this study provide support for engaging in moderate physical activity during pregnancy, seemingly without posing a risk of neonatal complications, it is imperative to approach these results with considerable caution, recognizing the imperative for further research in this area. To date, this is the first in-depth systematic review of the variables analysed. Therefore, high-quality studies are required to shed additional light on the relationship between exercise during pregnancy in terms of several fetal parameters.

For this reason, it is crucial to emphasize the value of exercise during pregnancy to reverse the physiological effects of physical inactivity in new generations. These findings can encourage pregnant women to maintain or achieve a minimum level of daily activity to improve neonatal health outcomes.

## 5. Conclusions

The quality of evidence from randomized controlled trials showed that prenatal physical activity could reduce the risk for NICU admissions compared with control groups. Additionally, PA during pregnancy contributes to better Apgar 1 and 5 scores among healthy pregnant women.

## Figures and Tables

**Figure 1 jpm-14-00006-f001:**
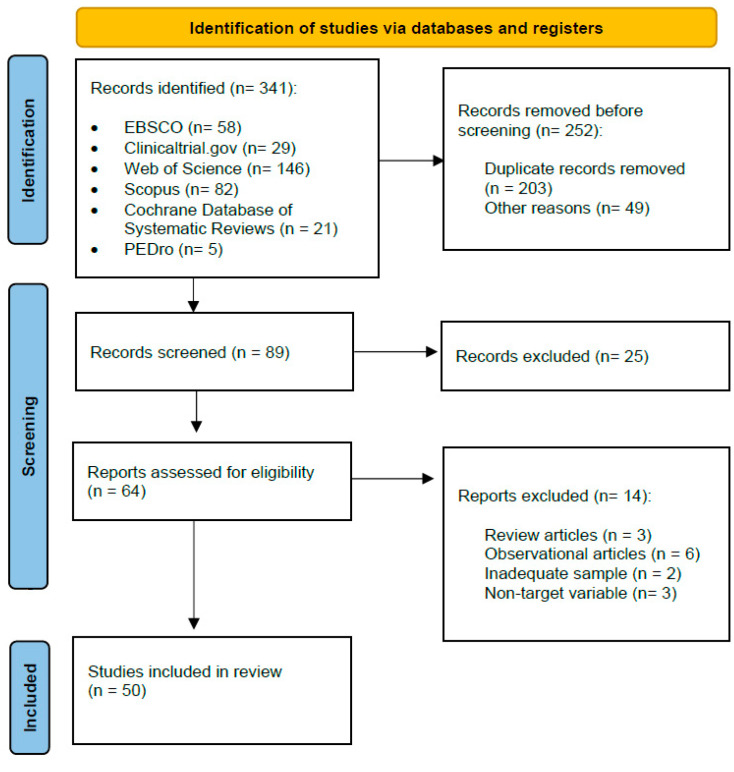
Flow chart of the review process.

**Figure 3 jpm-14-00006-f003:**
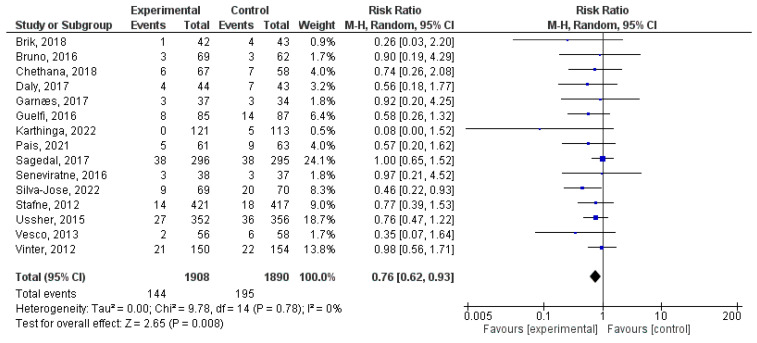
Forest plot of the effect of exercise during pregnancy on NICU admissions [5,43,44,46,48,50,53,55,60,69,71,73,75,76,77].

**Figure 4 jpm-14-00006-f004:**
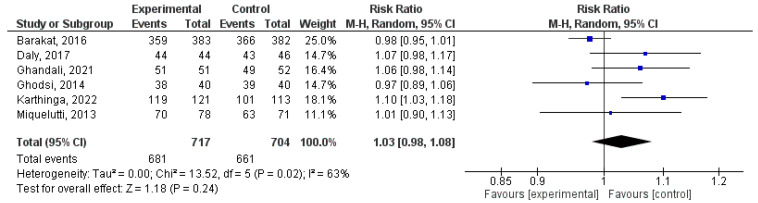
Forest plot of the effect of exercise during pregnancy on Apgar 1 scores > 7 [40,48,51,52,55,57].

**Figure 5 jpm-14-00006-f005:**
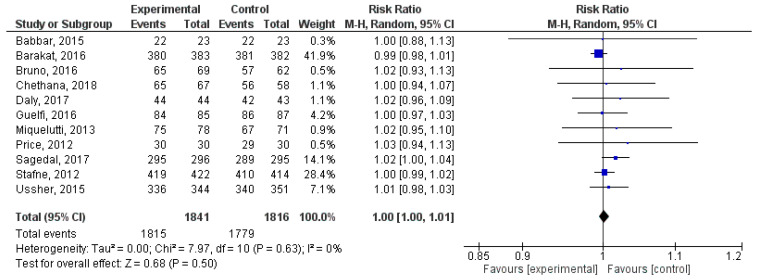
Forest plot of the effect of exercise during pregnancy on Apgar 5 scores > 7 [33,40,44,46,48,53,57,67,69,73,75].

**Figure 6 jpm-14-00006-f006:**
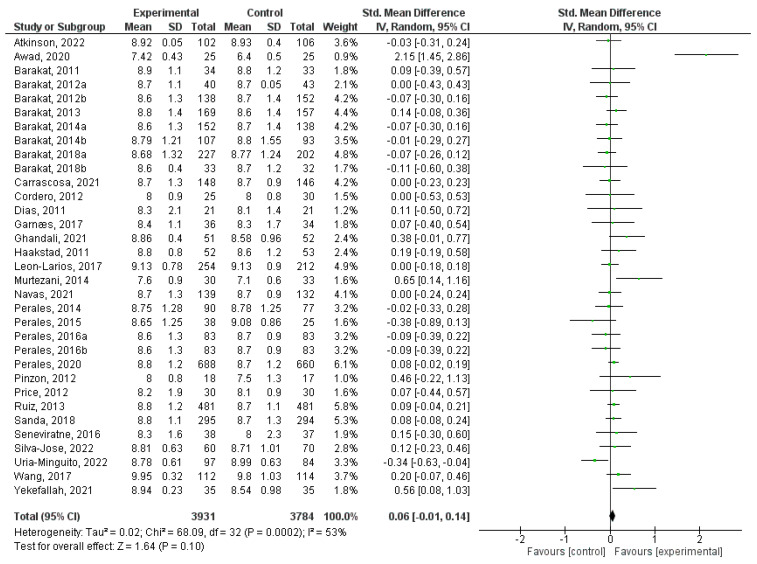
Forest plot of the effect of exercise during pregnancy on Apgar 1 scores [5,31,32,34,35,36,37,38,39,41,42,45,47,49,50,51,54,56,58,59,61,62,63,64,65,66,67,68,71,72,74,78,79].

**Figure 7 jpm-14-00006-f007:**
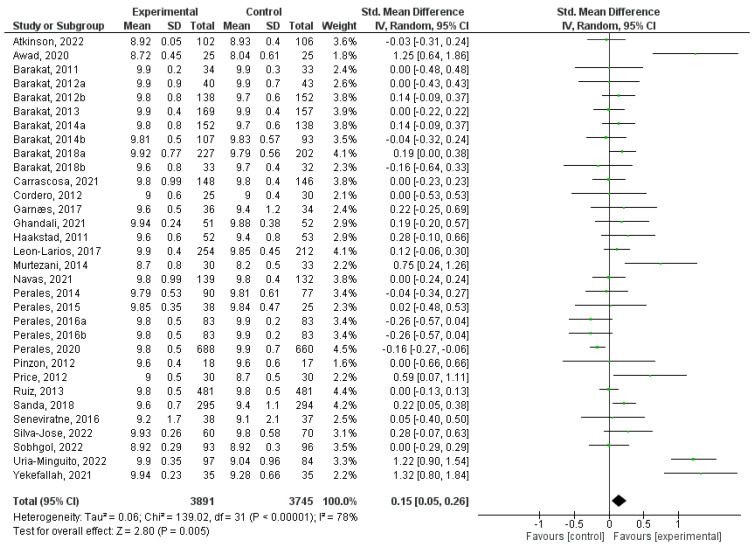
Forest plot of the effect of exercise during pregnancy on Apgar 5 scores [5,31,32,34,35,36,37,38,39,41,42,45,47,49,50,51,54,56,58,59,61,62,63,64,65,66,67,68,71,72,74,79].

## Data Availability

Not applicable.

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
