# Peer review of "Influence of Physical Activity during Pregnancy on Neonatal Complications: Systematic Review and Meta-Analysis"

_jpm, 2023, doi:10.3390/jpm14010006_

Round 1

Reviewer 1 Report

Comments and Suggestions for Authors

Dear Cristina Silva-Jose and colleagues, Thank you for the submission of your well written and interesting systematic review and meta-analysis for consideration for publication in Journal of Personalized Medicine. However, this reviewer finds that some issues could be addressed in order to even more increase the quality of the paper. Please see the specific comments to the authors:

The rationale for the review in the context of existing knowledge is not sufficiently described in the Introduction section. Therefore, I suggest the authors to add somewhere in Introduction something like this "Hypothalamic-pituitary-adrenal activation throughout pregnancy is linked to fetal programming (reference: Bolten M, Nast I, Skrundz M, et al. Prenatal programming of emotion regulation: neonatal reactivity as a differential susceptibility factor moderating the outcome of prenatal cortisol levels. J Psychosom Res. 2013 Oct;75(4):351-7.). Furthermore, high anxiety in mothers, autonomously of the baby's birth situation and the moment of evaluation, comprises a possible risk factor for the child's development, even in the fetal period (reference: Correia LL, Linhares MB. Maternal anxiety in the pre- and postnatal period: a literature review. Rev Lat Am Enfermagem. 2007 Jul-Aug;15(4):677-83.). Moreover, association between the time spent in physical activity and symptoms of depression and anxiety in antenatal period was previously demonstrated (reference: Petrovic D, Perovic M, Lazovic B, Pantic I. Association between walking, dysphoric mood and anxiety in late pregnancy: A cross-sectional study. Psychiatry Res. 2016 Dec 30;246:360-363.)." Otherwise, Manuscript is well written, with inclusion and exclusion criteria clearly specified, search strategies for all databases, registers and websites are well presented, all outcomes for which data were sought are listed and defined, the methods used to assess risk of bias in the included studies are specified, results are well presented and discussion is balanced.

Author Response

Dear Reviewer 1, thank you very much for your work and effort in reviewing our manuscript.

Based on your corrections, we have put them into action. We have added the proposed sentences in the introduction, updating the bibliographic references, to provide more quality to the manuscript.

  • "Hypothalamic-pituitary-adrenal activation throughout pregnancy is linked to fetal programming (reference: Bolten M, Nast I, Skrundz M, et al. Prenatal programming of emotion regulation: neonatal reactivity as a differential susceptibility factor moderating the outcome of prenatal cortisol levels. J Psychosom Res. 2013 Oct;75(4):351-7.). LINES 30-31
  • Furthermore, high anxiety in mothers, autonomously of the baby's birth situation and the moment of evaluation, comprises a possible risk factor for the child's development, even in the fetal period (reference: Correia LL, Linhares MB. Maternal anxiety in the pre- and postnatal period: a literature review. Rev Lat Am Enfermagem. 2007 Jul-Aug;15(4):677-83.). LINES 46-48

Moreover, association between the time spent in physical activity and symptoms of depression and anxiety in antenatal period was previously demonstrated (reference: Petrovic D, Perovic M, Lazovic B, Pantic I. Association between walking, dysphoric mood and anxiety in late pregnancy: A cross-sectional study. Psychiatry Res. 2016 Dec 30;246:360-363.)."  LINES 64-66

Reviewer 2 Report

Comments and Suggestions for Authors

Dear authors, I would first like to congratulate you on the work done. Prior the publication, there are some minor issues that should be resolved: 

Introduction

Line 45: Change ‘physical active’ to ‘physical activity’

Line 50: I would put the reference at the end of the sentence

Line 53: There are brackets missing for the reference.

Line 54: NICU admission FOR INFANTS of women… I do know that it is self-explanatory, but I would rather have the complete sentence here.

Methods:

Rephrase the ‘Population’ section, as you included the studies in which the mentioned exclusion and inclusion criteria were used.

For ‘Comparison’ section: ‘Not practicing PA’ was I believe, not being included in the intervention? Meaning that the pregnant women may have been active, just did not receive the intervention? I am right? Or were the intervention groups compared exclusively with sedentary women?

Results:

‘Republic of Kosovo’, I would advise the authors to put ‘Kosovo’ and then the explanation in the footnote: In accordance with the declaration 1244 of the UN.

Author Response

Dear Reviewer 2, thank you very much for your work and effort in reviewing our manuscript.

Based on your corrections, we have put them into action. We have added in the We have modified the proposed suggestions point by point.

Introduction

Line 45: Change ‘physical active’ to ‘physical activity’. Done, line 49

Line 50: I would put the reference at the end of the sentence Changed line 54

Line 53: There are brackets missing for the reference. Changed line 57

Line 54: NICU admission FOR INFANTS of women… I do know that it is self-explanatory, but I would rather have the complete sentence here. Changed lines 57-58

Methods:

Rephrase the ‘Population’ section, as you included the studies in which the mentioned exclusion and inclusion criteria were used. Done 87-90

For ‘Comparison’ section: ‘Not practicing PA’ was I believe, not being included in the intervention? Meaning that the pregnant women may have been active, just did not receive the intervention? I am right? Or were the intervention groups compared exclusively with sedentary women?

As they are all randomized clinical trials, the comparison was made with the control groups of the studies, which followed routine health care, without any established physical exercise program. In the same way we have reformulated the phrase to clarify the meaning. Lines 99-101

Results:                                                         

‘Republic of Kosovo’, I would advise the authors to put ‘Kosovo’ and then the explanation in the footnote: In accordance with the declaration 1244 of the UN.  Done, line 247

Reviewer 3 Report

Comments and Suggestions for Authors

The SR and MA are good but the writing style needs improvement. The Introduction is rambling and poorly focused for the reader ; try a Table to present the risks as listed.

Do not write using our / you use many phases that are not really scientific style 'at this point'; in this sense; on the other hand; foir this reason; 

Comments on the Quality of English Language

This needs moderate revision.

Author Response

Dear Reviewer 1, thank you very much for your work and effort in reviewing our manuscript.

The SR and MA are good but the writing style needs improvement. The Introduction is rambling and poorly focused for the reader ; try a Table to present the risks as listed.

Based on your corrections, we have put them into action. We have added greater clarity to the introduction by adding relevant data to put the reader in the situation.

  • "Hypothalamic-pituitary-adrenal activation throughout pregnancy is linked to fetal programming (reference: Bolten M, Nast I, Skrundz M, et al. Prenatal programming of emotion regulation: neonatal reactivity as a differential susceptibility factor moderating the outcome of prenatal cortisol levels. J Psychosom Res. 2013 Oct;75(4):351-7.). LINES 30-31
  • Furthermore, high anxiety in mothers, autonomously of the baby's birth situation and the moment of evaluation, comprises a possible risk factor for the child's development, even in the fetal period (reference: Correia LL, Linhares MB. Maternal anxiety in the pre- and postnatal period: a literature review. Rev Lat Am Enfermagem. 2007 Jul-Aug;15(4):677-83.). LINES 46-48

  • Moreover, association between the time spent in physical activity and symptoms of depression and anxiety in antenatal period was previously demonstrated (reference: Petrovic D, Perovic M, Lazovic B, Pantic I. Association between walking, dysphoric mood and anxiety in late pregnancy: A cross-sectional study. Psychiatry Res. 2016 Dec 30;246:360-363.)."  LINES 64-66

Do not write using our / you use many phases that are not really scientific style 'at this point'; in this sense; on the other hand; foir this reason; 

The entire text has been reviewed and the scientific style has been revised. Likewise, the English writing has been reviewed by a native specialist. These are some examples of that:

  • It was chosen not to exclude any study from the analyses. Line 160
  • In addition, an attempt was made to analyze data from other present neonatal birth outcomes such as… Line 328
  • study findings must be carefully and collectively weighed to formulate unique decisions lines 357-358
  • When it comes to physical exercise programs. Lines 361